# Predictive Value of Heat-Shock Protein Gene Expression on Severe Neonatal Hypoxic-Ischemic Encephalopathy

**DOI:** 10.3390/diagnostics12040981

**Published:** 2022-04-13

**Authors:** Yu-Mi Seo, Seok Hwang-Bo, Soo-Ah Im, Myungshin Kim, Young-Ah Youn

**Affiliations:** 1Department of Pediatrics, Seoul St. Mary’s Hospital, College of Medicine, The Catholic University of Korea, Seoul 06591, Korea; seoyum1130@naver.com (Y.-M.S.); hbs6144@naver.com (S.H.-B.); 2Department of Radiology, Seoul St. Mary’s Hospital, College of Medicine, The Catholic University of Korea, Seoul 06591, Korea; saim@catholic.ac.kr; 3Department of Laboratory Medicine, Seoul St. Mary’s Hospital, College of Medicine, The Catholic University of Korea, Seoul 06591, Korea; microkim@catholic.ac.kr

**Keywords:** biomarker, genes, heat-shock protein, hypoxic ischemic encephalopathy, seizure

## Abstract

This study aims to evaluate significant gene expression in severe hypoxic ischemic encephalopathy (HIE) in newborns, which can be used as a predictable measure for high-risk HIE infants. The study prospectively recruited 77 inborn near-term or term HIE newborns between January 2018 and December 2020. We measured six different genes within 6 h of life among the HIE infants and compared the gene levels between the mild- and severe-HIE groups. Among these, 64 HIE infants (83.1%) did not receive therapeutic hypothermia (TH) because they were categorized as mild HIE, and the 13 remaining (16.9%) infants were categorized as ≥ moderate-HIE group and received TH. More abnormal MRI findings, seizure, and use of anti-convulsant were more found in the ≥ moderate = HIE group along with longer mechanical ventilation days and hospitalization. Heat-shock protein 70 family 1 A (HSPA1A) and serpin family H member 1 (SERPINH1) genes, which encode heat-shock protein (HSP) 70 and 47, respectively, were significantly elevated in the ≥ moderate-HIE, seizure, and abnormal MRI groups. HSP 70 and 47 were significantly elevated in the severe-HIE group, possibly playing protective roles in inhibiting exacerbated neuroinflammation and maintaining a cellular homeostasis. At 18–24 months, ≥ moderate-HIE group manifested a significant language delay.

## 1. Introduction

While the incidence of HIE in developed countries is still estimated to be as high as 1.5 per 1000 live births [1,2], the molecular mechanisms and the pathway of brain injury in infants with hypoxic ischemic encephalopathy (HIE) remain largely unknown despite the progressive advances in neonatal care. Especially, the prognosis of HIE followed by intrapartum hypoxia often results an inevitable brain injury, which results in high (15 to 25%) mortality and severe morbidity, with an incidence of 25% of permanent neurological sequelae among the survived infants [3,4].

To minimize the life-long disability or mortality, therapeutic hypothermia (TH) is performed on HIE infants with moderate to severe HIE before 6 h of age. Despite the therapeutic success of hypothermia, moderate to severely affected newborns continue to have a 30–70% risk of death or disability [5,6], which requires further studies about adjunctive therapies for additional neuroprotective treatment. Ischemic injury causes changes at the cellular level due to the effects of systemic inflammation. Hypoxia and ischemia increase ROS production and cause cell damage. Perinatal HIE occurs due to the failure of the compensatory mechanism for such damage [7,8]. Further studies involving the underlying mechanisms of HIE injury at birth are needed to develop early predictive biomarkers. [9,10,11,12]. The availability of predictive markers of neuronal injury that correlate with disease severity and neurodevelopmental disability would facilitate a more targeted therapeutic approach. Recently, the role of heat-shock response in HIE patients is discussed in many studies [13,14,15,16,17]. Heat-shock proteins (HSPs) are a family of endogenous proteins that play key roles in protein homeostasis, acting as chaperones after acute insult [18]. They protect cells by guiding the degradation of irreversibly damaged proteins. Mostly, they suppress misfolding of proteins and aggregation of misfolded toxic proteins. Usually, HSPs are expressed at low levels under normal physiological conditions [19]. The role of these proteins is to prevent abnormal changes under stress [20]. HSPs respond to a variety of brain injuries [21], and their expression could rapidly increase in response to different acute brain injuries, such as ischemic stroke [18]. By doing so, they modulate neuronal survival and neuroinflammation after acute hypoxic ischemic encephalopathy and intervene in disease process signaling. In this prospective study, we aimed to examine significant key genes in HIE infants and explored the relationship of key genes in the severity of brain injury. We additionally searched to determine if the potential genes were correlated with abnormal neurodevelopmental outcomes at 12–18 months of age in HIE infants.

## 2. Materials and Methods

### 2.1. Subject Selection

This prospective cohort pilot study included all infants who were diagnosed with HIE, including those who were treated with TH (therapeutic hypothermia) in ≥ moderate stage of encephalopathic term or late preterm infants (≥35 weeks of gestation with birth weights ≥2000 g) between January 2018 and December 2020 at Seoul St. Mary’s Hospital, College of Medicine, Catholic University of Korea. All HIE infants were recruited for the initial blood sample for evaluation of unique gene expression profile. We collected 2 mL of blood (venous or arterial) from the recruited infants within 6 h of birth and prior to initiation of cooling therapy. The data were prospectively analyzed.

### 2.2. Exclusion Criteria

We excluded HIE infants with birth weights ≤2000 g and gestational age ≤34 weeks to minimize preterm related confounding factors in this HIE analysis. Those with major congenital abnormalities, syndromes, or metabolic diseases were also excluded in this study.

### 2.3. HIE Management for TH

Inclusion criteria were all infants who experienced acute perinatal events and were diagnosed with HIE. When ≥ moderate stage of HIE was diagnosed within 6 h of birth, TH treatment was performed. The enrolled infants fulfilled one of the two parameters as previously described in the CoolCap, NICHD, European Trial, and TOBY trials [5,6,22,23]. Cerebral function monitoring (CFM) (CFM, Natus Medical Inc., Seattle, WA, USA) or video EEG was started as early as possible to detect any possible electrographic seizures. Infants presenting with clinical seizures or abnormal aEEGs were considered ≥ moderate encephalopathy, which was used to determine the HIE eligibility for TH treatment. The evidence of moderate or severe encephalopathy was distinguished using the Sarnat and Sarnat clinical stages [24]. Seizures were clinically diagnosed (by experienced neonatologists or neurologist) as a paroxysmal alteration in motor function and occasional autonomic function; this included clonic, tonic, and “subtle” seizure manifestations [22]. Infants were randomly assigned to whole-body cooling (core esophageal temperature kept at 33.5 °C for 72 h).

#### 2.3.1. RRNA Extraction, Alignment, and Next-Generation Sequencing

Blood specimens (2 mL) were collected within 6 h from an indwelling umbilical arterial or venous line. Samples were refrigerated immediately and processed within 24 h. RNA was extracted using RNeasy^®^ Mini Kit (Qiagen, Hilden, Germany).

The blood was gently mixed with 1.4 mL RNA stabilizing solution (PreAnalytiX BD/QIAgen) and subsequently stored in a −80 °C freezer until analysis. We extracted total RNA from whole blood according to the manufacturer’s instructions and removed ribosomal RNA and globin mRNA from 4 μg of total RNA. RNA quality was assessed by analysis of rRNA band integrity on an Agilent RNA 6000 Nano kit (Agilent Technologies, Santa Clara, CA, USA). Ahead of cDNA library construction, 1 μg of total RNA and magnetic beads with Oligo (dT) were used to enrich poly (A) mRNA from it. Then, the purified mRNAs were disrupted into short fragments, and the double-stranded cDNAs were immediately synthesized. The cDNAs was subjected to end-repair, poly (A) addition, and connected with sequencing adapters using the TruSeq Stranded mRNA sample prep Kit (Illumina, San Diego, CA, USA). The final library sizes and qualities were evaluated electrophoretically with an Agilent High Sensitivity DNA kit (Agilent Technologies, Santa Clara, CA, USA). Subsequently, the library was sequenced using an Illumina NovaSeq6000 sequencer (Illumina, San Diego, CA, USA).

#### 2.3.2. Gene Expression Analysis by Real-Time Quantitative RT-PCR (RT-qPCR)

The candidate genes were assessed within 6 h of life in HIE infants to observe whether a surge in specific gene levels can stratify the severity of HIE and a unique blood gene expression profile at birth may indicate as a synergic biomarker for severe HIE. To confirm the gene expression level, we selected nine genes that revealed significant difference between two groups: defensin alpha 3 (DEFA3) (Hs00414018_m1), Fos proto-oncogene, AP-1 transcription factor subunit (FOS) (Hs00170630_m1), heat-shock protein family A (Hsp70) member 1A (HSPA1A) (Hs00359163_s1), interleukin 1 beta (IL1B) (Hs01555410_m1), orosomucoid 1 (ORM1) (Hs01590790_g1), oncostatin M (OSM) (Hs00171165_m1), serpin family H member 1 (SERPINH1) (Hs001060397_g1), transferrin (TF) (Hs00169070_m1), and zinc finger DHHC-type containing 19 (ZDHHC19) (Hs00376116_m1). cDNA was prepared using Transcriptor First strand cDNA Synthesis kit (Roche, Swiss, Basel) and GeneAmp PCR system 9700 (Applied biosystems, Foster City, CA, USA) according to manufacturers’ instruction. The gene expression levels were calculated by 2^−ΔCt^ method with GAPDH (Hs99999905_m1) as reference gene. All experiments were triplicated independently.

### 2.4. Imaging Methods

Brain MRI (MRI) with MR diffusion was performed in all HIE infants (at least within 10 days of life) after they were extubated from the ventilator or stabilized for the imaging study. One independent specialized radiologist who was masked to treatment and outcomes of the infants reviewed the brain MRI images for quality and acquired lesions.

### 2.5. Classification of Brain MRIs

Magnetic resonance imaging was performed mean 7 days of life after TH infants were stabilized, rewarmed, and extubated from the ventilator. The MRI was categorized according to patterns of structural injury according to the NICHD pattern for brain injury by: score of 0 for normal MRI; 1A for minimal cerebral lesions only; 1B for more extensive cerebral lesions without basal ganglia and thalamus (BGT) or posterior limb of internal capsule (PLIC) or anterior limb of internal capsule (ALIC) involvement and no area of watershed infarction; 2A for any BGT, PLIC, or ALIC involvement or watershed infarction without any cerebral lesions; 2B for any BGT, PLIC, or ALIC involvement or watershed infarction with additional cerebral lesions; and 3 for cerebral hemispheric devastation [25].

### 2.6. Neurodevelopmental (ND) Assessment

At 18–24 months, the TH infants returning for follow-up evaluations completed the cognitive, language, and motor composites of the Bayley Scales of Infant and Toddler Development III and were evaluated by certified examiners. Children were considered as developmental delay (DD) if scores were below 84. If the scores were above 85, they were considered to be normal in their developmental stage.

The study was conducted in accordance with the Declaration of Helsinki by the Ethics Committees of Seoul St. Mary’s Hospital, and the written informed consent was obtained from the parents and/or legal guardians of all the subjects. The experimental protocol was approved by the Ethics Committees of Seoul St. Mary’s Hospital (# KC19TNSI0174). The work was funded by the National Research Foundation of Korea (NRF) granted by the Korean government of Ministry of Science (2019R1G1A1002198).

### 2.7. Statistical Analysis

Continuous variables are expressed as the means ± standard deviations and were compared with Student’s *t*-test. Continuous variables are displayed as median with inter-quantile range when the variables are not normally distributed and compared with Wilcoxon rank sum test.

All statistical analyses were two-tailed, with statistical significance defined as values of *p* < 0.05. All statistical analyses were performed with SPSS, version 15.0 (Statistical Package for the Social Sciences, SPSS-PC Inc., Chicago, IL, USA).

## 3. Results

The study prospectively recruited 77 inborn near-term or term HIE newborns between January 2018 and December 2020 (Figure 1).

All infants were categorized by severity of HIE: mild-HIE infants (64; 83.1%) were those who did not receive TH, and the remaining 13 (16.9%) infants were categorized in the moderate-to-severe-HIE group and received TH. Descriptive clinical characteristics of HIE newborns are presented in Table 1.

Mostly baseline clinical characteristics were similar between the mild- and moderate-to-severe (≥ moderate) HIE groups. The emergent C/S delivery was significantly more found in the moderate-HIE group along with significantly lower Apgar scores at 1 and 5 min (*p* < 0.05). Meconium aspiration syndrome (MAS) suggesting fetal distress was significantly higher, and the muscle enzymes, such as lactate dehydrogenase (LDH) and creatine phosphokinase (CPK), were significantly higher in the ≥ moderate-HIE group. In regard to hospital outcomes, clinical seizures were significantly associated with moderate-HIE group, and the first and second AED uses were significantly higher (*p* < 0.001). Consequently, abnormal MRI findings were more found in the ≥ moderate-HIE group. The mechanical ventilation days and hospitalization were significantly longer in the ≥ moderate-HIE group. To note, HSPA1A (heat-shock protein 70 family 1 A) and SERPINH1 (serpin family H member 1) genes were significantly elevated in the ≥ moderate-HIE group (Table 1). Regarding seizure, the baseline clinical characteristics were similar between the infants who experienced seizure compared to no seizure HIE group. Infants who had seizure manifested significantly lower Apgar scores at 1 and 5 min (*p* < 0.05). Initial LDH and CPK were significantly higher with more anti-epileptic drug (AED) use in the seizure group. The mortality was significantly higher in the seizure group. Similarly, HSPA1A and SERPINH1 genes were significantly elevated in the seizure HIE group (Table 2).

When grouped according to MRI findings, clinical seizures and AED (especially phenobarbital) use was significantly higher in the abnormal MRI group. HSPA1A and SERPINH1 genes were consistently significantly elevated in the abnormal brain MRI group, with FOS gene significantly decreased (*p* < 0.05) (Table 3).

At 18–24 months, the moderate-to-severe-HIE infants manifested a significant delay in the language composite of the Bayley Scales of Infant and Toddler Development III. Infants at risk were also higher in the moderate-to-severe-HIE infants (Table 4).

## 4. Discussion

Most centers rely on clinical neurological examination in adjuvant to Sarnat staging [24] to identify ≥ moderate-HIE infants needing TH. However, clinical examination is subjective and often confounded by various clinical factors. Gunn et al. reported that initial clinical examination was recently shown to be a less reliable predictor of outcome in infants needing therapeutic hypothermia [26]. The emerging simplified amplitude integrated electroencephalogram tool to detect subtle seizures was recently demonstrated by us and others to have limited early predictive value in infants treated with hypothermia [27,28]. In our study, two key genes related to ≥ moderate-HIE were HSPA1A and SERPINH1 genes, which were significantly elevated in comparison to mild-HIE infants. The infants with significantly elevated HSPA1A and SERPINH1 genes, which encode HSP 70 and 47, experienced more clinical seizure and more abnormal brain MRI findings. After all, the genes measured within 72 h were consistently elevated in severe HIE cases.

The numbers after the HSP generally indicate the size of the protein, which is numbered according to protein molecular weights. HSP plays a role in maintaining intracellular homeostasis by processing damaged proteins in various stressful situations (heat, hypoxia, cold, UV radiation, etc.), thereby called “stress proteins.” When cells are under stressful conditions, HSP is more greatly produced through modulation of signal transduction pathways to protect damage to intracellular proteins through chaperon function.

HSPs are present in the cytosol, mitochondria, endoplasmic reticulum, and nucleus. The 70-kDa heat-shock protein (HSP 70) is one of the major stress proteins induced by cerebral ischemia [29,30]. Mainly, they suppress misfolding of proteins and aggregation of misfolded toxic proteins. Usually, HSPs are expressed at low levels under normal physiological conditions [19], but they are elevated under stress because the role of these proteins is to prevent abnormal changes in the composition of cells [20].

In an animal study, HSP 70 widely spread to increase in areas of the cerebral cortex, white matter, and hippocampus after reperfusion of ischemia, which suggests the overall protection from injuries and its role in maintaining the homeostasis [31]. The increased expression of HSPs has been shown to be protective in many cultured cells and animal tissues functioning as one of the molecular chaperones to imbalance the protein homeostasis network. Their function is linked to protein folding and assembly and protein degradation [32] to prevent inappropriate protein aggregation and to mediate transport of immature proteins to the target organelles for final packaging, degradation, or repair [33]. In addition to protecting cells from proteotoxic stress by preventing protein aggregation and promoting degradation of misfolded proteins, HSP 70 also modulates inflammatory pathways. Yu et al. found that the over-expression of HSP 70 suppressed neuroinflammation and further decreased the basal level of inflammatory cytokines, such as TNF-α and IL-1. On the other hand, the inhibition of HSP 70 exacerbated neuroinflammation in astrocytes [34]. Therefore, HSP 70 is an important modulator in regulating neuroinflammation by astrocytes activation. Its role as a key regulatory protein is reported in the process of neuroinflammation during brain hypoxia and neurodegenerative disorders, such as Parkinson’s and Alzheimer’s diseases [35]. The significant elevation of HSP 70 not only plays as a modulator but also is protective in severe-HIE infants against inflammatory cytokines, minimizing secondary damage and regulating cellular homeostasis. In cases of cancer cells [36], the overexpression of HSP 70 indicates survival rate from stress situations, such as anticancer drugs.

Several studies have raised the possibility that HSP 70 may be involved in various aspects of the immune system [37]. The overexpression of HSP 70 is capable of also down-regulating signal transduction, altering enzymatic activities such as diminishing protein kinase activity and thereby maintaining a cellular homeostasis. They are reported to play crucial roles in the survival of organisms because they are ubiquitously present in cells under both normal and pathological conditions [33].

In our study, SERPINH1 (serpin family H member 1) was also significantly elevated in more severe-HIE infants, those with more clinical seizure, those who needed TH, and the more severe brain MRI group. SERPINH1 is also a heat-shock protein 47 and less-known than HSP 70 for its brain-cell-protective effect. Heat-shock protein 47 (HSP 47) is a member of the serpin superfamily of serine proteinase inhibitors and a molecular chaperone involved in the maturation of collagen molecules [38]. HSP 47 is encoded by the SERPINH1 gene, which is located on chromosome 11q13.5, one of the most frequently amplified regions in human cancer. Thereby, overexpression of HSP 47 might result in the continuing growth of fibrosis, resulting in tumor angiogenesis, uninhibited cell growth, and cancer [39]. SERPINH1 encodes HSP 47 and is lesser known than HSP 70 for its brain-cell-protective effect [38]. The vascular fibrosis inflammation occurs in this reperfusion period, which aggravates more neuronal damage. The increased SERPINH1 gene in our severe-HIE infants may as well reflect more neuronal damage associated to over-deposit of proteins and fibrosis in reperfusion period [40]. Elevation of this specific gene is observed in patients with acute ischemic injury. From this, it can be considered as a biomarker that can estimate the prognosis of HIE patients. Specifically, HSP 47 plays an important role in collagen-binding protein and may result overproduction of procollagen and fibrosis in the human body. Miyamura et al. used a small-molecule HSP 47 inhibitor to an antifibrotic effect by suppressing the overexpression of collagen and migration of fibroblasts [41]. TH is administered to minimize secondary inflammation during reperfusion period. The vascular fibrosis inflammation occurs in this reperfusion period, which aggravates more neuronal damage. Kelly et al. explained in their study that the large amounts of fibrosis and extracellular matrix (ECM) protein deposition occurring during the reperfusion period can impede neuronal repair [40]. Therefore, the increased serpine H1 gene in our severe-HIE infants may as well reflect more neuronal damage associated to over deposit of proteins and fibrosis in reperfusion period.

In our study, the c-fos mRNA was significantly elevated in HIE infants with normal brain MRI findings. Welsh et al. found that the expression of c-fos mRNA was produced even a minor degree of localized trauma; however, intense expression was reported following middle cerebral artery occlusion throughout the ipsilateral cerebral cortex in an experimental rat model [42]. Minor degree of localized trauma was sufficient to trigger expression of c-fos, which may be well detected in the mild-HIE patients in our study. c-fos mRNA is additionally reported to be triggered by a spreading depression, which is characterized by recurrent waves of neuronal depolarization by a transient influx of calcium [43]. Expression ofc-fos is regulated, in part, by calcium influx through voltage-sensitive channels [44]. Thus, induced calcium influx after brain injury could express the widespread expression of c-fos gene.

The key genes measured within 72 h of age in HIE newborns did not predict abnormal neurodevelopmental outcomes at 12 to 18 months. It could be the effect of TH or the brain-cell-protective effect of HPS. Earlier diagnosis and intervention of HIE are critical to lessen the neurological sequel and to decrease the neonatal mortality [45]. Even earlier TH intervention has shown reduced neuronal loss and improved survival and brain functional outcome afterwards. The sooner cooling can be initiated after injury, the more likely it is to be successful, as many human experimental studies additionally manifested that the sooner the TH is started, the better the therapeutic effect results [46].

Some studies described elevation of LDH and CPK enzymes following the hypoxic insult [47]. These enzymes were significantly elevated in this study as well. Although significant increases were observed, it is difficult to see the two enzyme levels as tissue-specific. Another biomarker is a serum brain-specific protein, including S100B in HIE newborns, which is also elevated in infants with brain damage diagnosed by MRI [48]. It is expected that the gene expression examined in this study will be more suitable to specifically reflect the effects of HIE-induced acute brain injury and the severity. Therefore, we considered that the two genes mentioned in our results would be suitable biomarkers as likely indicators to be used to assess the initial severity of HIE.

This study had some limitations. Several factors may have contributed to a potential selection bias in our analysis: First, we had a relatively small sample size in study group due to appropriate blood sampling in infants; second, hidden disabilities may subsequently have become apparent later after 18 months of age, and some infants might have important developmental lags that were not classified as impairments.

In addition, brain MRI performed to evaluate the degree of acute ischemic injury was taken around 7 days of age. Many infants receiving therapeutic hypothermia are intubated, and brain MRI is done after extubation. Therefore, the mean brain MRI examination time was 7 days after birth in a extubated state. In order to evaluate the degree of acute ischemic injury, there is a point that the time of imaging is delayed compared to that of blood samples taken within 6 h before hypothermia therapy was performed. In some of the literature, diffusion-weighted brain MRI within 10 days is reported to reflect early ischemic brain injury in association with perinatal HIE accurately [49]. The imaging suitably reflects birth asphyxia. With this, we sought to assess the effects of ischemic injury and to identify associations with specific gene expression observed in patients in the early stages of the injury.

After acute ischemic event, the elevation of those specific genes might reflect as potential prognostic biomarkers in the severity of HIE infants. These potential validated genes as biomarkers can optimize neurotherapeutic trials by more accurately risk stratifying patients at entry, serving as surrogate outcomes in the future. The risk stratification to guide additional neuroprotective therapies in the future may improve our early ability to detect and predict outcomes. Further studies on HIF1 [50] as a gene that is transcribed in hypoxic stress and encodes HSP-90 may probe its association with HIE infants in the future.

## 5. Conclusions

Heat-shock protein 70 family 1 A (HSPA1A) and serpin family H member 1 (SERPINH1) genes, which encode heat-shock protein (HSP) 70 and 47, respectively, were significantly elevated in the ≥ moderate-HIE, seizure, and abnormal MRI groups. HSP 70 and 47 were significantly elevated in the severe-HIE group, possibly playing protective roles in inhibiting exacerbated neuroinflammation and maintaining a cellular homeostasis. At 18–24 months, ≥ moderate-HIE group manifested a significant language delay.

## Figures and Tables

**Figure 1 diagnostics-12-00981-f001:**
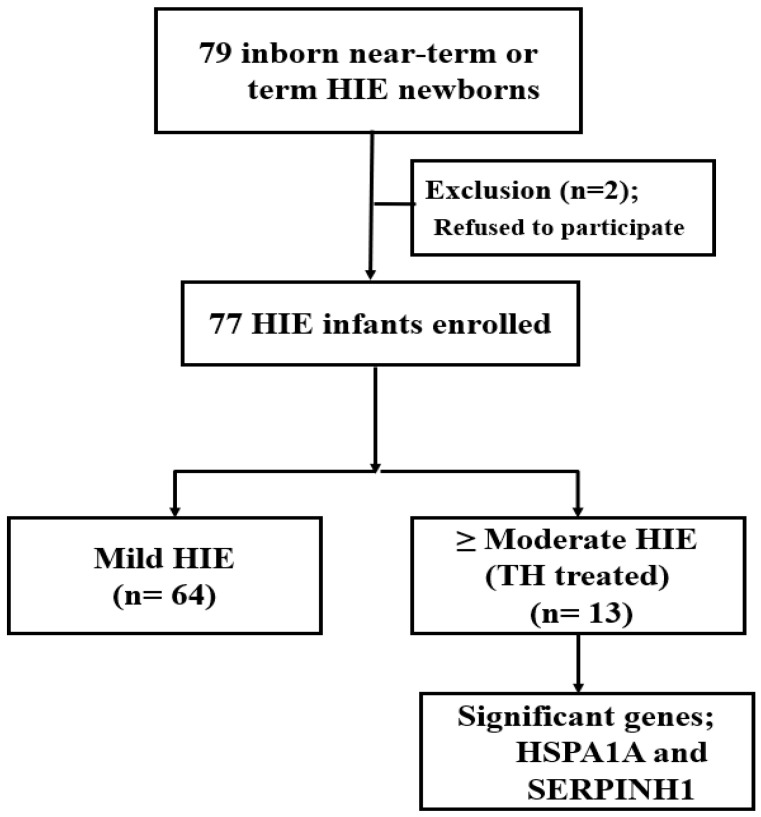
Study Flow chart; heat-shock protein genes predicting severe neonatal hypoxic-ischemic encephalopathy. Abbreviations: HIE, hypoxic ischemic encephalopathy; TH, therapeutic hypothermia; Hsp70, heat-shock protein family A; SERPINH1, serpin family H member 1.

**Table 1 diagnostics-12-00981-t001:** Clinical characteristics of HIE infants (n = 77).

	Mild HIE (n = 64)	≥Moderate HIE (TH, n = 13)	*p*-Value
Gestational age, week	37.882 ± 2.228	37.956 ± 1.679	0.713
Birth weight, kg	3.026 ± 0.533	2.915 ± 0.723	0.678
Emergent delivery	25 (39.1)	9 (69.2)	0.046
Apgar score at 1 min	7.063 ± 2.137	4.385 ± 2.785	0.002
Apgar score at 5 min	8.391 ± 1.857	5.692 ± 3.225	0.002
Outborn, n (%)			
Male, n (%)	40 (62.5)	7 (53.8)	0.560
MAS, n (%)	4 (6.3)	5 (38.5)	0.006
SGA, n (%)	7 (10.9)	3 (23.1)	0.359
Fetal heart deceleration	7 (10.9)	4 (30.8)	0.083
Initial PH < 7.0	2 (3.1)	1 (7.7)	0.430
LDH (initial)	1115.125 ± 1115.292	2192.615 ± 1588.452	<0.001
CPK (initial)	649.922 ± 549.473	1357.154 ± 1444.528	0.022
Sarnat Stage on Day 1			N/A
Stage 2	1 (1.6)	3 (23.1)	
Stage 3	2 (3.1)	7 (53.8)	
Sarnat Stage on Day 4			N/A
Stage 2	0 (0)	0 (0)	
Stage 3	1 (1.6)	4 (30.8)	
Clinical seizure	6 (9.4)	9 (69.1)	<0.001
AED, phenobarbital	7 (10.9)	10 (76.9)	<0.001
AED, keppra	3 (4.7)	4 (30.8)	0.014
AED, phenytoin	0	0	N/A
Ventilator care, days	3.09 ± 4.166	28.08 ± 48.683	<0.001
Abnormal MRI	12 (18.8)	10 (76.9)	0.001
Developmental delay	16 (25)	5 (38.5)	0.325
Hospital days	15.25 ± 12.80	42.46 ± 44.70	<0.001
Death	3 (4.7)	3 (23.1)	0.057
Genes			
GAPDH	23.509 ± 1.678	22.680 ± 1.089	0.142
DEFA3	26.397 ± 46.884	17.684 ± 12.209	0.174
FOS	0.399 ± 0.639	0.377 ± 0.375	0.751
HSPA1A	0.303 ± 0.852	2.093 ± 5.608	<0.001
IL-1b	0.049 ± 0.050	0.031 ± 0.026	0.479
ORM1	0.172 ± 0.191	0.215 ± 0.248	0.860
OSM	0.005 ± 0.006	0.003 ± 0.003	0.146
SERPINH1	0.012 ± 0.040	0.065 ± 0.163	0.006
TF	0.008 ± 0.023	0.002 ± 0.003	0.812
ZDHHC19	0.001 ± 0.001	0.001 ± 0.001	0.548

Abbreviations: HIE, hypoxic ischemic encephalopathy; TH, therapeutic hypothermia; MAS, meconium aspiration syndrome; SGA; small for gestational age; BE, base excess; LDH, lactate dehydrogenase; CPK, creatine phosphokinase, AED, anti-epileptic drug; GAPDH, glyceraldehyde-3-phosphate dehydrogenase; DEFA3, defensin alpha 3; FOS (proto-oncogene, AP-1 transcription factor subunit); HSPA1A, Hsp70 heat-shock protein family member 1A; IL1B, interleukin 1 beta; ORM1, orosomucoid 1; OSM, oncostatin M; SERPINH1, serpin family H member 1; TF, transferrin; ZDHHC19, zinc finger DHHC-type containing 19.

**Table 2 diagnostics-12-00981-t002:** Significant clinical findings in clinical seizure infants (n = 77).

	Normal (n = 62)	Seizure (n = 15)	*p*-Value
Gestational age, week	38.009 ± 2.214	37.419 ± 2.192	0.207
Birth weight, kg	3.054 ± 0.541	2.814 ± 0.642	0.179
Emergent delivery	24 (38.7)	10 (66.7)	0.050
Apgar score at 1 min	7.06 ± 2.137	4.38 ± 2.785	0.040
Apgar score at 5 min	8.39 ± 1.857	5.69 ± 3.225	0.010
Outborn, n (%)			
Male, n (%)	39 (62.9)	8 (53.3)	0.495
MAS, n (%)	6 (9.7)	3 (20.0)	0.366
SGA, n (%)	7 (11.3)	3 (20.0)	0.399
Fetal heart deceleration	7 (11.3)	4 (26.7)	0.210
Emergent call	13 (21.0)	6 (40.0)	0.180
Initial PH < 7.0	2 (3.2)	1 (6.7)	0.483
Initial BE	−6.589 ± −7.104	−10.433 ± −8.827	0.059
LDH (initial)	1005.145 ± 426.325	2503.533 ± 2510.961	<0.001
CPK (initial)	635.952 ± 552.600	1320.600 ± 1341.018	0.008
Sarnat Stage on Day 1			N/A
Stage 2	1 (1.6)	3 (20.0)	
Stage 3	3 (4.8)	6 (40.0)	
Sarnat Stage on Day 4			N/A
Stage 2	1 (1.6)	3 (20.0)	
Stage 3	2 (3.2)	3 (20.0)	
AED, phenobarbital	2 (3.2)	15 (100)	<0.001
AED, keppra	1 (1.6)	6 (40.0)	<0.001
AED, phenytoin	0	0	N/A
Ventilator care, days	5.129 ± 16.423	16.333 ± 36.164	<0.001
Developmental delay	14 (22.7)	7 (46.7)	0.102
Hospital days	17.419 ± 19.065	29.867 ± 36.109	0.124
Mortality	2 (3.2)	4 (26.7)	0.012
Genes			
GAPDH	23.465 ± 1.758	22.974 ± 0.735	0.643
DEFA3	26.141 ± 47.230	19.905 ± 18.027	0.598
FOS	0.407 ± 0.648	0.349 ± 0.356	0.471
HSPA1A	0.251 ± 0.727	2.068 ± 5.248	0.001
IL-1b	0.049 ± 0.051	0.031 ± 0.027	0.258
ORM1	0.161 ± 0.176	0.254 ± 0.275	0.375
OSM	0.005 ± 0.006	0.003 ± 0.003	0.361
SERPINH1	0.011 ± 0.036	0.062 ± 0.155	0.025
TF	0.008 ± 0.023	0.006 ± 0.010	0.450
ZDHHC19	0.001 ± 0.001	0.001 ± 0.001	0.225

Abbreviations: MAS, meconium aspiration syndrome; SGA; small for gestational age; BE, base excess; LDH, lactate dehydrogenase; CPK, creatine phosphokinase, AED, anti-epileptic drug; GAPDH, glyceraldehyde-3-phosphate dehydrogenase; DEFA3, defensin alpha 3; FOS (proto-oncogene, AP-1 transcription factor subunit); HSPA1A, Hsp70 heat-shock protein family member 1A; IL1B, interleukin 1 beta; ORM1, orosomucoid 1; OSM, oncostatin M; SERPINH1, serpin family H member 1; TF, transferrin; ZDHHC19, zinc finger DHHC-type containing 19.

**Table 3 diagnostics-12-00981-t003:** Significant clinical findings in abnormal brain MRI (n = 77).

	Normal (n = 56)	Abnormal MRI (n = 21)	*p*-Value
Gestational age, week	37.88 ± 2.23	37.96 ± 1.68	0.877
Birth weight, kg	3.03 ± 0.53	2.92 ± 0.72	0.991
Emergent delivery	22 (39.3)	12 (57.1)	0.160
Apgar score at 1 min	7.06 ± 2.137	4.38 ± 2.785	0.012
Apgar score at 5 min	8.39 ± 1.857	5.69 ± 3.225	0.056
Outborn, n (%)			
Male, n (%)	37 (66.1)	10 (47.6)	0.139
MAS, n (%)	7(12.5)	2 (9.5)	1.000
SGA, n (%)	8 (14.3)	2 (9.5)	0.719
Fetal heart deceleration	7 (12.5)	4 (19.0)	0.479
Emergent call	11 (19.6)	8 (38.1)	0.094
Initial PH < 7.0	3 (5.4)	0 (0)	0.558
Initial BE	6.122 ± 6.304	13.323 ± 10.345	0.740
LDH (initial)	1115.125 ± 1115.292	2192.615 ± 1588.452	0.063
CPK (initial)	649.922 ± 549.473	1357.154 ± 1444.528	0.277
Sarnat Stage on Day 1			N/A
Stage 2	2 (3.6)	2 (9.5)	
Stage 3	4 (7.1)	5 (23.8)	
Clinical seizure	4 (7.1)	11 (52.4)	<0.001
AED, phenobarbital	5 (8.9)	12 (57.1)	<0.001
AED, keppra	2 (3.6)	5 (23.8)	0.014
AED, phenytoin	0	0	N/A
Ventilator care, days	4.875 ± 17.024	13.810 ± 30.919	0.001
Developmental delay	11 (19.6)	10 (47.6)	0.014
Hospital days	15.25 ± 12.80	42.46 ± 44.70	0.001
Death	5 (8.9)	1 (4.8)	1.000
Genes			
GAPDH	23.435 ± 1.803	23.194 ± 0.986	0.855
DEFA3	22.989 ± 42.572	30.091 ± 45.078	0.507
FOS	0.424 ± 0.670	0.320 ± 0.359	0.042
HSPA1A	0.361 ± 0.914	1.257 ± 4.470	0.058
IL-1b	0.048 ± 0.051	0.040 ± 0.036	0.714
ORM1	0.174 ± 0.201	0.193 ± 0.201	0.656
OSM	0.005 ± 0.006	0.003 ± 0.003	0.123
SERPINH1	0.015 ± 0.043	0.038 ± 0.130	0.020
TF	0.009 ± 0.024	0.003 ± 0.006	0.850
ZDHHC19	0.001 ± 0.001	0.001 ± 0.001	0.210

Abbreviations: MRI, magnetic resonance imaging; MAS, meconium aspiration syndrome; SGA; small for gestational age; BE, base excess; LDH, lactate dehydrogenase; CPK, creatine phosphokinase, AED, anti-epileptic drug; GAPDH, glyceraldehyde-3-phosphate dehydrogenase; DEFA3, defensin alpha 3; FOS (proto-oncogene, AP-1 transcription factor subunit); HSPA1A, Hsp70 heat-shock protein family member 1A; IL1B, interleukin 1 beta; ORM1, orosomucoid 1; OSM, oncostatin M; SERPINH1, serpin family H member 1; TF, transferrin; ZDHHC19, zinc finger DHHC-type containing 19.

**Table 4 diagnostics-12-00981-t004:** Neurodevelopmental outcomes of HIE on Bayley Scales of Infant and Toddler Development III at corrected 18 months (n = 77).

	Mild HIE (n = 60)	Moderate-to-Severe HIE (TH-Treated) (n = 10)	*p*-Value
Score, mean ±SD			
Cognitive	101.47 ± 13.04	102.7 ± 16.09	0.789
Language	100.3 ± 11.14	83.6 ± 13.30	0.041
Motor	98.52 ± 16.19	95.3 ± 14.97	0.559
At risk, n (%)			
Cognitive delay	1 (1.7%)	0 (0.0%)	1.000 ^a^
Language delay	0 (0.0%)	1 (10.0%)	0.304 ^a^
Motor delay	2 (3.3%)	1 (10.0%)	0.904 ^a^

Abbreviations: HIE, hypoxic ischemic encephalopathy; TH, therapeutic hypothermia. ^a^ Children were considered to be at risk if their scores were >2 standard deviations below the test mean (scores of <70).

## Data Availability

Not applicable.

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
