# Peer review of "Predictive Value of Heat-Shock Protein Gene Expression on Severe Neonatal Hypoxic-Ischemic Encephalopathy"

_diagnostics, 2022, doi:10.3390/diagnostics12040981_

Round 1

Reviewer 1 Report

The authors submitted a manuscript investigating a topic of great clinical and medico-legal interest.

The manuscript must extensively improve in both editing and English.

Referencing is often poor and generally way too dated, considering the issue under investigation and the great efforts devoted by the scientific community. This issue has to be addressed by the authors.
In particular:

  • Line 45-47: '...further studies involving the underlying mechanisms to develop 45 early...': this statement deserve a wider referencing than a single 2012 paper. Novel approaches should be taken into consideration (starting from 10.1007/s11010-021-04253-8 10.3390/diagnostics12010214 10.3390/metabo10040141 10.1515/cclm-2019-0725);
  • Introduction does not provide any background about the study rationale and previous insight on HSP and perinatal asphyxia and neonatal hypoxic ischemic encephalopathy. Literature should be double check and appropriately cited in order to introduce the reader into the topic (see 10.1007/s12192-017-0782-0  10.1159/000370031 10.1038/sj.jcbfm.9600080).
  • Line 231 and 233: the authors are citing papers 'recently' published although they refer to aged papers. Should not the authors provide truly recent papers, 'recently' need to be removed from text.

Introduction:

Line 55-63: as previously reviewed the introduction does not bring the reader inside the topic, the need for further investigation and the way this would be pursued. Some of the general contents included in the first part of the discussion might be moved here.
Possibly, the major issue regarding the paper is whether the genes investigated were chose by chance, by evidence from previous papers/data, or they emerge during a general unknown approach. This should be clearly addressed in all the manuscript sections.

Materials and Methods:

Line 127-142: the two sections are a bit redundant and might be condensed in a shorter, single, paragraph.

Discussion:

Line 239-244: The paragraph is hard to read. please reorder concept in a more logical way.

Line 286-287: the two sentences are badly linked as the reader suddenly come across TH. Please provide a better link.

In the reviewer opinion a significant result is completely ignored by the authors in the discussion. LDH, and sometimes CPK, actually provide the same information than the HSP genes investigated. Apparently a potential practical application of HSP gene expression would be importantly limited by a cheaper, faster and easier biochemical analysis. This issue should be appropriately faced by the authors in the discussion paragraph (and consequently in the limitations one).

Limitation:

Sample size issue should be mentioned in a more direct way as the difference between the two groups is far to be slight.

Another important limitation that should be mentioned is that MRI are performed 7 days after birth offering an ex post perspective on the effects rather than the cause.

Author Response

<1st reviewer>

Response to Dear reviewer 1,

Thank you for your valuable comments. We’ve revised as you suggested as the following;

The authors submitted a manuscript investigating a topic of great clinical and medico-legal interest.

The manuscript must extensively improve in both editing and English.

Referencing is often poor and generally way too dated, considering the issue under investigation and the great efforts devoted by the scientific community. This issue has to be addressed by the authors.
In particular:

  • Line 45-47: '...further studies involving the underlying mechanisms to develop 45 early...': this statement deserve a wider referencing than a single 2012 paper. Novel approaches should be taken into consideration (starting from 10.1007/s11010-021-04253-8 10.3390/diagnostics12010214 10.3390/metabo10040141 10.1515/cclm-2019-0725);
  • Based on your advice, I have added a recent article related to this subject (Lafemina et al., 2006; Matara et al., 2022; Debuf et al., 2021; Bersani et al., 2019).

  • Introduction does not provide any background about the study rationale and previous insight on HSP and perinatal asphyxia and neonatal hypoxic ischemic encephalopathy. Literature should be double check and appropriately cited in order to introduce the reader into the topic (see 10.1007/s12192-017-0782-0 10.1159/000370031 10.1038/sj.jcbfm.9600080).
  • We added more description about HSP and HIE. Please see the attached paragraph in line 53-61 in Discussion section.

Recently, the role of heat-shock response in HIE patients is discussed in many studies [11-13]. Heat-shock proteins (HSPs) are a family of endogenous proteins which play key roles in protein homeostasis acting as chaperones after acute insult [14]. They protect cells by guiding the degradation of irreversibly damaged proteins. Mainly, they suppress misfolding of proteins and aggregation of misfolded toxic proteins. Usually, HSPs are expressed at low levels, and under normal physiological conditions [15]. The role of these proteins is preventing changes in the composition of cells under stress [13]. HSPs respond to a variety of brain injuries [16], and their expression could rapidly increase in response to different acute brain injuries, such as ischemic stroke [14]. By doing so, they modulate neuronal survival, neuroinflammation after acute hypoxic ischemic encephalopathy and intervene disease process signaling.

  • Line 231 and 233: the authors are citing papers 'recently' published although they refer to aged papers. Should not the authors provide truly recent papers, 'recently' need to be removed from text.
  • As you pointed out, the phrase was removed. Thank you.

Introduction:

Line 55-63: as previously reviewed the introduction does not bring the reader inside the topic, the need for further investigation and the way this would be pursued. Some of the general contents included in the first part of the discussion might be moved here.
Possibly, the major issue regarding the paper is whether the genes investigated were chose by chance, by evidence from previous papers/data, or they emerge during a general unknown approach. This should be clearly addressed in all the manuscript sections
.

  • This study compared the gene expression levels according to the severity of HIE patients and evaluated the clinical course and neurodevelopmental prognosis based on the information presented in several previous papers (C.Y.Liu et al., 2021; Ponnusamy V. et al., 2019; Shen G. et al., 2020; Ma Q. et al., 2017; Bustelo M. et al.,2020; Bersani et al., 2019). We targeted a number of genes in relation to HIE in this study. Only two genes (HSPA1A and SERPINH1) which encode HSP 70 and 47 came out to be significantly related to severe HIE neonates which reflects ischemic injury in brain.

Materials and Methods:

Line 127-142: the two sections are a bit redundant and might be condensed in a shorter, single, paragraph.

  • Edited for conciseness based on your advice.
  • Neurodevelopmental (ND) assessment : At 18-24 months, the TH infants returned for follow-up evaluations completed the Bayley Scales of Infant and Toddler Development III. Children were considered as developmental delay (DD) if scores were below 84. If the scores were above 85, they were considered to be normal in their developmental stage.
  • The experimental protocol was approved by the Ethics Committees of Seoul St. Mary's Hospital (# KC19TNSI0174). The work was funded by the National Research Foundation of Korea (NRF) granted by the Korean government of Ministry of Science (2019R1G1A1002198).

Discussion:

Line 239-244: The paragraph is hard to read. please reorder concept in a more logical way.

  • As you pointed out, we try to correct the description of HSP. Please refer to line 263-269 in discussion section.

Therefore, HSP 70 is an important modulator in regulating neuroinflammation by astrocytes activation. It’s role as a key regulatory protein is reported in the process of neuroinflammation during brain hypoxia and neurodegenerative disorders such as Parkinson's and  Alzheimer's disease [30]. The significant elevation of HSP 70 not only play as a modulator, but also play protectively in severe HIE infants against inflammatory cytokines to minimize secondary damage and regulating cellular homeostasis. In cases of cancer cells [31], the over expression of HSP 70 indicates survival rate from stress situations such as anticancer drugs.

Line 286-287: the two sentences are badly linked as the reader suddenly come across TH. Please provide a better link.

  • As you suggested, we try to make clear that HSPA1A and SERPINH1 genes, encode HSP 70 and 47, more detailed information regard to HSP 70 and HSP 47 and their key roles in HIE were described and added. Please refer to line 239-242 in discussion section.

In our study, two key genes related to moderate to severe HIE were HSPA1A and SERPINH1 genes ,which encode HSP 70 and 47, respectively were significantly elevated in comaprison to mild HIE infants. The infants with signitificantly elevated HSPA1A and SERPINH1 genes, which encode HSP 70 and 47, experienced more clinical seizure and more abnormal MRI findings.

  • We added more description SERPINH1 (HSP 47). Please refer to line 282-287 in discussion section.

SERPINH1 which encode heat-shock protein 47 and less known than HSP 70 for its brain cell protective effect [33]. The vascular fibrosis inflammation occurs in this reperfusion period, which aggravates more neuronal damage. the increased SERPINH1 gene in our severe HIE infants may as well reflect more neuronal damage associated to over deposit of proteins and fibrosis in reperfusion period [35]. Elevation of this specific gene is observed in patients with acute ischemic injury. From this, it can be considered as a biomarker that can estimate the prognosis of HIE patients.

In the reviewer opinion a significant result is completely ignored by the authors in the discussion. LDH, and sometimes CPK, actually provide the same information than the HSP genes investigated. Apparently a potential practical application of HSP gene expression would be importantly limited by a cheaper, faster and easier biochemical analysis. This issue should be appropriately faced by the authors in the discussion paragraph (and consequently in the limitations one).

  • As you mentioned, we added about non specific roles of LDH and CPK enzyme elevation in relation to HIE; Please see the attached paragraph in Line 312-319 in Discussion section.

Some studies described elevation of these enzymes following the hypoxic insult [42]. Although not statistically significant, both LDH and CPK were significantly elevated in this study as well. But these LDH and CPK enzymes are not specific because they are released from various tissues, not mainly originated brain. To date, some biomarkers do not originate in the brain, but have been used as predictive indicators of some outcomes, such as long-term impairments in neurological development and death [43]. Another biomarker is a serum brain-specific protein, including S100B in HIE newborns, which is elevated in infants with brains damage diagnosed by MRI [44]. It is expected that the gene expression examined in this study will be more suitable to specifically reflect the effects of HIE-induced acute brain injury and to evaluate the severity.   

Limitation:

Sample size issue should be mentioned in a more direct way as the difference between the two groups is far to be slight. .

  • Among HIE neonates enrolled in our study; only those who are categorized as ≥ moderate HIE received TH (therapeutic hypothermia). The mild HIE neonates did not receive TH. Therefore, our severity category is based on TH which means more severe HIE. TH were classified as moderate to severe HIE for analysis. Among them, as you pointed out, this study has limitations in size of our patient number, but since it was prospective enrollment (not intended recruitment of patients). We mainly aimed to investigate the neurodevelopmental outcomes of moderate to severe HIE in this study. The contents of that part are more meaningful and more research is needed in the future.

Another important limitation that should be mentioned is that MRI are performed 7 days after birth offering an ex post perspective on the effects rather than the cause.

  • As mentioned in the review, brain MRI was performed around 7 days of age. Many infants were intubated during hypothermia and the brain MRI was performed after extubation. Therefore, the mean time for brain MRI was about 7 days of age (after extubation). Diffusion-weighted Brain MRI within 10 days is reported to reflect early ischemic brain injury in association to perinatal HIE accurately (Agut et al., 2014). The imaging suitably reflects birth asphyxia (K.Murphy et al., 2017). With this, we sought to assess the effects of ischemic injury and to identify associations with specific gene expression observed in patients in the early stages of the injury.

Again, thank you for your valuable comments.

Sincerely,

Dr. Youn, YoungAh

Reviewer 2 Report

I appreciate the author presenting this research article. My comments are as follows

  1. Introduction:The importance of the early detection of gene expression should be mentioned in
  2. The drawback of current study is the study design is not well. Two groups including mild HIE and moderate HIE with TH are inadequate . It is very difficulty to know the positive or negative effects due to  moderate HIE itself or or TH. This made the results are not easy to read. 

Author Response

<2nd reviewer>

Response to Dear reviewer 1,

Thank you for your valuable comments. We’ve revised as you suggested as the following;

I appreciate the author presenting this research article. My comments are as follows

  1. Introduction: The importance of the early detection of gene expression should be mentioned in

  • We added more description about HIE in association of genes which encode heat shock proteins(HSPs). Please see the attached paragraph in Line 48-64 in Discussion section

Further studies involving the underlying mechanisms to develop early predictive measures such as biomarkers that identify the extent of neurological injury at birth or thereafter were studied [7-10]. The availability of predictive markers of neuronal injury that correlate with disease severity and neurodevelopmental disability would facilitate a more targeted therapeutic approach.

Recently, the role of heat-shock response in HIE patients is discussed in many studies [11-13]. Heat-shock proteins (HSPs) are a family of endogenous proteins which play key roles in protein homeostasis acting as chaperones after acute insult [14]. They protect cells by guiding the degradation of irreversibly damaged proteins. Mainly, they suppress misfolding of proteins and aggregation of misfolded toxic proteins. Usually, HSPs are expressed at low levels, and under normal physiological conditions [15]. The role of these proteins is preventing changes in the composition of cells under stress [13]. HSPs respond to a variety of brain injuries [16], and their expression could rapidly increase in response to different acute brain injuries, such as ischemic stroke [14]. By doing so, they modulate neuronal survival, neuroinflammation after acute hypoxic ischemic encephalopathy and intervene disease process signaling.

In this study, we aimed to examine an array of key gene expression involving HSPs and compared the severity of brain injury in relation to significant gene levels. The goal of this prospective study was to explore the relationship of key genes which might predict the severity and outcomes of HIE infants.

  1. The drawback of current study is the study design is not well. Two groups including mild HIE and moderate HIE with TH are inadequate . It is very difficulty to know the positive or negative effects due to moderate HIE itself or of TH. This made the results are not easy to read.

  • As you pointed out, it is expected that there will be some confusion in interpreting the effects of HIE itself or of TH. First of all, in this study, HIE was divided into two groups. Among newborns who received ischemic injury, those who did not receive TH were classified into a mild HIE group, and when TH was performed, they were classified and analyzed as a moderate to severe HIE group. In particular, the purpose of this study was to investigate the characteristics of gene expression measured in the acute phase to predict the severity of HIE infants.

Again, thank you for your valuable comments.

Sincerely,

Dr. Youn, YoungAh

Round 2

Reviewer 1 Report

The authors revised the manuscript working mainly on introduction and discussion sections. Furthermore they offered a rebuttal for several issue of the first revision.

Although improved, the manuscript has still major issues that needs to be addressed.

Referencing:

the papers 10.3390/metabo10040141 and 10.1007/s11010-021-04253-8 would improve the introduction and need to be cited among studies 'involving underlying mechanisms'.

Line 52-53: the authors refers to 'many studies' discussing HSP role in HIE although they cite just 3 papers 1 of which is a rat model of cerebral ischemia. Please improve referencing or correct the sentence.

Introduction:

Line 48-50: 'future studies...were studied' Please make the sentence less redundant.

Line 62-63: the authors indicate the goal of the study as the investigation of the relationship between key genes encoding HSPs and HIE. This is not correct as several genes among the ones investigated does not correlate to HSP. The issue regarding on which bases the authors selected the 9 genes to investigare still stand up. The rebuttal to this point may improve the manuscript quality.

Materials and methods:

line 113-115: profile or biomarker?it is one or the either, please address this point.

line 116: again, the sentence 'we selected 9 genes that revealed significant differences between two groups' comes up with a doubt in the reader mind. Do the authors select 9 genes among more or investigate only 9 genes? what is more if you selected among more (in that case the rest of the manuscript should be adapted) it is not true that all the 9 genes revealed significant differences (as per p value reported).

Discussion:

Line 312-313: the authors interpret as not statistical significant LDH and CPK, despite p-value similar, or sometimes better, to HSPA1A and SERPINH1. Also, the sentence opening the paragraph refers to 'these enzyme' without mentioning which one.
Limitation paragraph should include MRI issue.

Author Response

Response to Dear reviewer 1,

Thank you for your valuable comments. We’ve revised as you suggested as the following;

The authors revised the manuscript working mainly on introduction and discussion sections. Furthermore they offered a rebuttal for several issue of the first revision.

Although improved, the manuscript has still major issues that needs to be addressed.

Referencing:

the papers 10.3390/metabo10040141 and 10.1007/s11010-021-04253-8 would improve the introduction and need to be cited among studies 'involving underlying mechanisms'.

  • As you mentioned, the recent literature is explained as follows.
  • Ischemic injury causes changes at the cellular level due to the effects of systemic inflammation. Hypoxia and ischemia increase ROS production and cause cell damage. perinatal HIE occurs due to the failure of the compensatory mechanism for such damage [7,8].

Line 52-53: the authors refers to 'many studies' discussing HSP role in HIE although they cite just 3 papers 1 of which is a rat model of cerebral ischemia. Please improve referencing or correct the sentence.

  • Thanks for the editorial advice. With this in mind, we have added the following documents as references [10-12].
  • 2020 HSP 70 Induction Chaperonotherapy for Neuroprotection after Brain Injury
  • 2017 Comparison of the predictive value of prooxidant-antioxidant balance and HSPs
  • 2019 Association of Neonatal Asphyxia With Serum Levels of Heat Shock Protein 27

Introduction:

Line 48-50: 'future studies...were studied' Please make the sentence less redundant.

  • The sentences are modified into a simpler sentence as below.
  • LINE 51-53; Further studies involving the underlying mechanisms of HIE injury at birth are needed to develop early predictive biomarkers.

Line 62-63: the authors indicate the goal of the study as the investigation of the relationship between key genes encoding HSPs and HIE. This is not correct as several genes among the ones investigated does not correlate to HSP. The issue regarding on which bases the authors selected the 9 genes to investigare still stand up. The rebuttal to this point may improve the manuscript quality.

  • There was a series of key genes related to HIE in human studies. Among these, as you commented, some of the genes do not encode HSPs. Therefore, the paragraph was changed as below.
  • LINES 66-68; In this prospective study, we aimed to examine significant key genes in HIE infants and explored the relationship of key genes in the severity of brain injury. We additionally searched to determine if the potential genes were correlated with abnormal neurodevelopmental outcomes at 12–18 months of age in HIE infants.

Materials and methods:

line 113-115: profile or biomarker? it is one or the either, please address this point.

  • Thank you for taking a detailed look at our experiments and asking questions. After reviewing several papers, we selected 9 candidate genes (DEFA3, FOS, HSPA1A, IL1B, ORM1, OSM, SERPINH1, TF, ZDHHC19) which were analyzed and profiled by PCR in blood samples obtained from HIE infants within 6 hours of birth. As a result, we tried to check the possibility of application as a biomarker predicting the initial severity of HIE by selecting genes that are particularly highly expressed.

line 116: again, the sentence 'we selected 9 genes that revealed significant differences between two groups' comes up with a doubt in the reader mind. Do the authors select 9 genes among more or investigate only 9 genes? what is more if you selected among more (in that case the rest of the manuscript should be adapted) it is not true that all the 9 genes revealed significant differences (as per p value reported).

  • As you pointed out, I will modify the above sentence to make it clearer. As a result of the first test with a sample collected from the subject, other genes were also detected in addition to the genes shown in the results. Through data investigation, we selected 9 genes presumed to be related to ischemic injury which were analyzed between the two groups.(2020, Cengiz et al. Developmental differences in microglia morphology and gene expression; 2020, Montaldo P. et al. Transcriptomic profile of adverse neurodevelopmental outcomes after neonatal encephalopathy)

Discussion:

Line 312-313: the authors interpret as not statistical significant LDH and CPK, despite p-value similar, or sometimes better, to HSPA1A and SERPINH1. Also, the sentence opening the paragraph refers to 'these enzyme' without mentioning which one.
Limitation paragraph should include MRI issue.

  • Thanks for your comments on this part. On this occasion, we have once again carefully considered the consequences and implications of both enzymes and would like to comment on them.
  • Some studies described elevation of LDH and CPK enzymes following the hypoxic insult [47]. These enzymes were significantly elevated in this study as well. Although significant increases were observed, it is difficult to see the two enzyme levels as tissue-specific. Therefore, we considered that the two genes mentioned in our results would be suitable biomarkers as more likely indicators to be used to assess the initial severity of HIE.
  • In addition, brain MRI performed to evaluate the degree of acute ischemic injury was taken around 7 days of age. Many infants receiving therapeutic hypothermia are intubated and brain MRI is done after extubation. Therefore, the mean brain MRI examination time was 7 days after birth in a extubated state. In order to evaluate the degree of acute ischemic injury, there is a point that the time of imaging is delayed compared to that of blood samples taken within 6 hours before hypothermia therapy was performed. In some literature, diffusion-weighted Brain MRI within 10 days is reported to reflect early ischemic brain injury in association to perinatal HIE accurately (Agut et al., 2014). The imaging suitably reflects birth asphyxia (K.Murphy et al., 2017). With this, we sought to assess the effects of ischemic injury and to identify associations with specific gene expression observed in patients in the early stages of the injury.

Again, thank you for your valuable comments.

Sincerely,

Dr. Youn, YoungAh

Reviewer 2 Report

The revised manuscript has replied my comments partly. The drawback of current study is  only two groups (mild HIE and moderate HIE + HT) without moderate HIE group. Reject is my final decision.

Author Response

Response to Dear reviewer 2,

The revised manuscript has replied my comments partly. The drawback of current study is only two groups (mild HIE and moderate HIE + HT) without moderate HIE group. Reject is my final decision.

  • Although we thank you again for your review of this study, I am sorry for not fully satisfying your comments. If the answer to what was mentioned in the last review was partial, it is probably due to a lack of my understanding. This review has given you a chance to reconsider your advice.

As mentioned, in this study, the study subjects were divided into two groups (mild HIE and moderate HIE who were severe enough to receive therapeutic hypothermia(TH)). The intent to classify the group receiving therapeutic hypothermia was to distinguish the severity between the two groups. Receiving therapeutic hypothermia is indicative of the severity of the initial HIE condition. This point may be an area that can sufficiently confuse the interpretation of the results. This is because only infants with HIE of this severity experienced therapeutic hypothermia. Consequently, more severe HIE group who received TH manifested a significant language delay at corrected 18-24 months age (Table 4).

  • We hope that our reply answer your questions.

Again, thank you for your valuable comments.

Sincerely,

Dr. Youn, YoungAh

Round 3

Reviewer 1 Report

The authors revised the manuscript improving it both under ease of reading and overall scientific quality. The manuscript is now eligible for publication.